# Isolation and screening of fungi for enhanced agarwood formation in *Aquilaria sinensis* trees

**Chuang Liu**[1,2], **Guoying Zhou**[1,2], **Junang Liu**[1,2]*

1 College of Life Science and Technology, Central South University of Forestry and Technology, Changsha, Hunan, China, 2 Southern Plantation Forest Pest Prevention and Control State Forestry Administration Key Laboratory, Hunan Provincial Key Laboratory of Forest Pest Control, Economic Forest Cultivation and Protection Ministry of Education Key Laboratory, Changsha, Hunan, China

* kjc9620@163.com

**Data Availability Statement:** This manuscript's minimal data set is publicly available via Figshare at https://figshare.com/s/78ec5ddde0a6c424badb.

**Funding:** National Key R & D Program of China (2023YFD1401304-04 to J.L.) There was no additional external funding received for this study.

## Abstract

Agarwood is a resinous heartwood of *Aquilaria sinensis* that is formed in response to mechanical wounding. In the present study pre-treatment of *Aquilaria sinensis* was carried out, and then the dominant fungi were isolated and purified from the surface and electro-shock holes of trees. The isolated *Trichoderma sp.* and *Neurospora sp.* were then screened for resistance against benzyl acetone and then inoculated into healthy *Aquilaria sinensis* trees. After six months, the agarwood was collected for analysis. The chemical composition of incense was analyzed using gas chromatography-mass spectroscopy, and 82 chemical constituents were identified. Agarwood products formed by using *Trichoderma sp.* and *Neurospora sp.* consisted of 50.22% and 48.71% ether extracts, respectively, which surpassed the 10% threshold specified by the Chinese Pharmacopoeia. Similarly, relative aromatic contents in the two agarwood products were 30.1% and 32.86%, while proportions of sesquiterpene constituents were 10.21% and 11.19%, respectively. These two agarwood-specific chemical constituents accounted for a large proportion of the total chemical composition, which showed that the generated agarwood was of good quality. The results of the study revealed that both *Trichoderma sp.* and *Neurospora sp.* were able to effectively induce agarwood production in *Aquilaria sinensis* trees in 6 months. This study expands the library of fungi that promote the production of agarwood from *Aquilaria sinensis* trees.

## Introduction

*Aquilaria sinensis* (Lour.) Gilg is an evergreen tree that grows in subtropics and tropics, and is a unique species in China. The resin-containing wood of *Aquilaria sinensis* is known as agarwood [1]. When *Aquilaria sinensis* tree is subjected to natural or man-made injuries, such as slash, injury, disease, insects, lightning, etc., the resin is secreted, which keeps accumulating in the xylem over the years. Prolonged accumulation of resin leads to higher quality of agarwood [2]. In China, agarwood is considered a useful resource [3] with rich medicinal properties. Being a valuable resource, it has been exploited to the point of extinction, leading to shortage of agarwood supply. In 1987, *Aquilaria sinensis* was listed as a national rare and endangered

**Competing interests:** The authors have declared that no competing interests exist.

plant, requiring the third level of protection [4]. This plant was also included in the List of Wild Plants under State Key Protection in 1999 as a national key plant categorized under the second level of protection [5], and in the Atlas of Rare and Endangered Plants of Guangdong Province in 2003 [6]. The Pharmacopoeia of 2010 stipulates that *Aquilaria sinensis*, as the only source of authentic agarwood, is a national second-class endangered protected plant in China [7].

The naturally occurring agarwood has long outrun the market demand, and at the end of the 20[th] century, cultivation of *Aquilaria sinensis* started to become more and more common [8]. Currently, *Aquilaria sinensis* cultivation is mainly popular in Hainan and Guangdong provinces of China, and its cultivation scale is continuously expanding [9, 10]. In view of the current status of agarwood production in China, there is a need to develop standardized agarwood production technology using *Aquilaria sinensis*. Since 1930s, agarwood production by artificial inoculation of fungi have been explored by many research groups [11, 12]. Production of agarwood from *Aquilaria sinensis* can be facilitated by using a variety of fungi, but this process has been shown to be affected by different growth environments required for fungi [13]. In 1976, researchers at the Guangdong Institute of Botany first reported that fungal infection in *Aquilaria sinensis* resulted in agarwood formation [14]. In 1998, Qi et al. reported that *Menanotus flavolives* accelerated agarwood formation [15]. Similarly, Subeham et al. inoculated *Fusarium laseritum* into the holes in the trunk of *Aquilaria sinensis* and obtained agarwood one year later [16]. Another study demonstrated improved agarwood production from *Fusarium* infected *Aquilaria sinensis* [17]. In 2017, a study reported that *Lasiodiplodia theobromae* fungus promoted agarwood formation [18]. These investigations were mainly focused on the direct use of strains to promote the formation of agarwood in *Aquilaria sinensis* trees [19]. Pretreatment of *Aquilaria sinensis* trees have not been explored in any report before the artificial injections of fungi in *Aquilaria sinensis* trees to induce the formation of agarwood. Studies on endophytic fungi growing on medicinal plants have shown that endophytic fungi can not only synthesize their own active components, but also have the ability to promote the synthesis of active components by the host plant [20]. Fungi promoting the formation of agarwood are firstly tolerant to some extent to the active ingredients of agarwood [21]. Benzyl acetone is a plant antitoxin produced by *Aquilaria sinensis* wood tissues after fungal infection, which is absent in healthy *Aquilaria sinensis* trees themselves, and its formation may be related to the self-protection mechanism of wood tissue cells.The composition of agarwood is complex, consisting of three main components: sesquiterpenes, 2-phenylethylchromones, and aromatic compounds. Benzyl acetone is a representative compound of the aromatic content in agarwood [21].

To meet the demands of booming domestic *Aquilaria sinensis* cultivation industry, it is important to not only expand the library of fungi that promote agarwood formation, but also explore fast and effective agarwood formation techniques.The method presented here is able to narrow down the range of fungi that promote agarwood formation, and reduce the time and monetary cost of screening these fungi.

## Materials and methods

### Pre-treatment of *Aquilaria sinensis* trees

The test site was located in the State Forestry Farm of Chengmai County, Hainan Province (19˚40′ E, 110˚0′ N). Before fungal exposure, *Aquilaria sinensis* trees were subjected to fire and electric shock pretreatment (Fig 1).

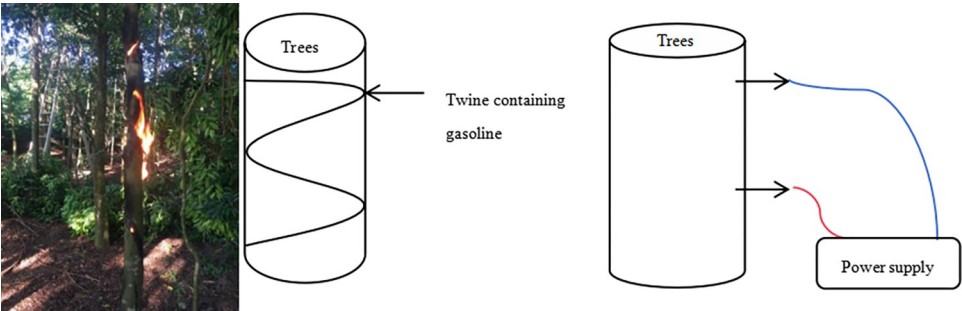

**Fig 1. Schematic diagram of burning method and electric shock.**

## Isolation, purification and identification of fungi

The surface of *Aquilaria sinensis* trees was covered with red and green fungal colonies after fire pretreatment (Fig 2). Subsequently, the bark carrying the fungal colonies was peeled off. The discolored wood of *Aquilaria sinensis* was collected after treatment by the electric shock pre-treatment for 3 months (S1 Fig). Fungal colonies collected from tree surface were inoculated onto potato dextrose agar (PDA) medium using a sterile scalpel. Collected discolored wood was cultured in situ using tissue blocks after tissue surface disinfection, and purification was continued until pure colonies of strains were isolated [22].

Purified fungal mycelium were picked and transferred to 1.5 ml centrifuge tube, and the fungal DNA was extracted by using Fungal DNA Extraction Kit, as per instructions of manufacturer. The extracted DNA was stored at -20˚C until further use. ITS1-4 primers were used for amplification of fungal DNA, and the extracted DNA samples were sent to Beijing Prime Biological Company for sequencing.

## Screening of benzyl acetone-tolerant fungi

Primary screening: The fungal strains obtained from isolation and purification were inoculated onto PDA medium and cultivated as test organisms. To screen the benzyl acetone-tolerant fungal strains, benzyl acetone was added to the medium at a concentration of 0.10% (v/v). The primary screening was carried out by fungal disc method. Round fungal discs were punched out from the PDA plate by using a 8 mm hole punch. The fungal disc was put into the center of the configured soil sedum medium plate containing benzyl acetone. As control, a fungal disc was put into the soil sedum medium without benzyl acetone. Both test and control experiment were conducted with three replicates. After incubation, the diameter of the bacterial circle (including the disc) was used

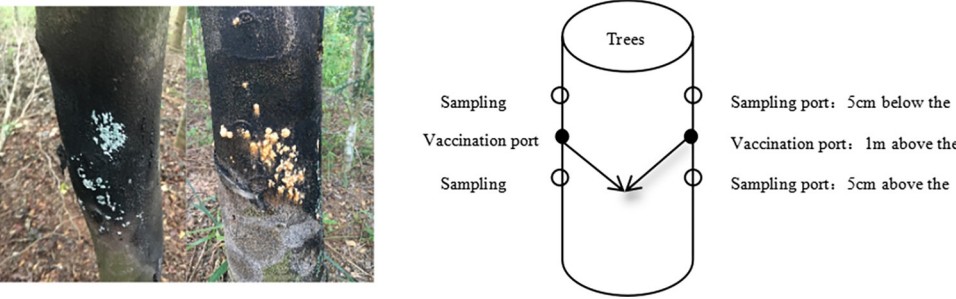

**Fig 2. Picture of the of the dominant colony on the surface of the tree, the dripping and sampling position of the fungal liquid.**

as a tolerance index. Bacterial growth inhibition rate was calculated by using the diameter of the bacterial circle, and the strains that showed the tolerance to benzyl acetone were selected for secondary screening [22]. The growth inhibition rate was calculated as follows:

Mycelial growth inhibition = (control colony diameter—treated colony diameter) / (control colony diameter– 8) x 100% [21]

Secondary screening: During secondary screening, experimental groups 1, 2 and 3 were set up by varying the benzyl acetone concentration in *Aquilaria sinensis* medium to 0.12%, 0.14%, and 0.16%, respectively. The fungal strains selected through primary screening were then transferred to the experimental plates as well as control plates (without benzyl acetone) in form of discs. After incubation, colony growth and diameter of the hyphae (including the fungus disc) were used as the indicators of benzyl acetone-tolerance. Fungal strains with stronger tolerance to benzyl acetone were selected to promote agarwood formation in *Aquilaria sinensis*.

## Artificial inoculation of fungi to induce the formation of agarwood in *Aquilaria sinensis*

The fungi *Trichoderma sp*. and *Neurospora sp*, which showed high tolerance to benzyl acetone, were selected and inoculated into liquid PDA media, separately. After incubation, fungal mycelia were filtered. The spore solutions were transferred to infusion bag, and then sterile water was added to obtain the final volume of 1000 ml. The two experimental groups related to two selected fungi were labelled as LV and H, respectively. In the control group, 1000 ml sterile water (without fungal spore liquid) was taken, and this group was labelled as CK. In this experiment, 5 replicates were set up in each group.

*Aquilaria sinensis* trees with a diameter of about 10–15 cm were selected, and a small hole of 7 mm (diagonally downward at 30 degrees) was symmetrically drilled at around 1 m above the ground, with a depth of 10 cm (Fig 2). These holes were used as the inoculation ports. On a sunny day, saps of experimental and control groups were injected separately into the trunk of selected trees. After 6 months of inoculation, wood samples were collected from above and below the inoculation ports of trees. After removing the white wood and decayed parts, the wood samples were ground into powder and stored at -80˚C.

## GC-MS detection of agarwood samples

"The agarwood" were extracted from the wood samples by using ultrasonic extraction in an ice-water bath. Extracted agarwood samples were then analyzed by GC-MS. Chromatographic conditions: the chromatographic column was a flexible quartz capillary column HP-5MS 5% Phenyl Methyl Siloxane (50 m×0.25 mm×0.25 μm); the heating procedure: the column temperature was 50˚C, held for 2 min, and then heated up to 310˚C at 5˚C/min, and held for 10 min; the temperature of the vapourization chamber was 250˚C; the carrier gas was high purity He (99.999%); carrier gas flow rate 1.0 mL/min; no splitting; solvent delay time: 5.0 min. Instrument model, Shimadzu QP2010 ultra.

Mass spectrometry conditions: electron bombardment (EI) ion source; electron energy 70 eV; ion source temperature 230˚C; quadrupole temperature 150˚C; interface temperature 280˚C; emission current 34.6 μA; multiplier voltage 1434 V; mass scanning range of 20~550 m/z [20].

## Results

### Isolation and identification of agarwood formation-inducing fungi

Total 26 fungal strains were isolated and purified from the surface of *Aquilaria sinensis* trees and discolored wood using the tissue block plate isolation method (S1 Fig). Using molecular

identification, the dominant fungal strains on the surface of *Aquilaria sinensis* were initially identified as *Trichoderma sp.* and *Neurospora sp.*, which were labelled as LV and H, respectively(S2–S4 Figs). The endophytic fungal strains isolated from the discolored wood section were initially identified as *Fusarium sp.* (16 strains), *Lasiodiplodia sp.* (3 strains), and *Talaromyces sp.* (5 strains).

## Screening of benzyl acetone-tolerant fungi

A certain extent, the isolated and purified fungal strains were screened for tolerance to benzyl acetone. All 26 fungal strains showed varying degrees of tolerance to benzyl acetone (S1 Table). Among these strains, 22 fungal strains showed highly significant differences, while 3 fungal strains showed moderately significant differences. One fungal strain showed no significant difference compared to control, which indicated that this strain was less affected by the presence of benzyl acetone in the medium and was more tolerant. Among them, the growth inhibition rates of LV and H differed significantly from those of other fungi. The inhibition rate of one fungal strain, CX3-2 (2), was as high as 100%, which indicated that the fungus was weakly tolerant to benzyl acetone. After primary screening, the strains with inhibition rate of 50% or less were selected for secondary screening. In secondary screening experiment, growth of control and experimental groups of each selected fungus differed significantly. In general, growth inhibition rate increased with the increase in benzyl acetone concentration in the medium (S2 Table). Compared to control, LV did not show any significant difference in growth at benzyl acetone concentration of 0.12%, but showed significant difference at benzyl acetone concentration of 0.14%. Compared to other fungi, LV and H showed decreasing differences in tolerance as the volume fraction of benzyl acetone in media increased.

## Promotion of agarwood formation in *Aquilaria sinensis* by artificial inoculation of benzyl acetone-tolerant fungi

Growth of *Aquilaria sinensis* and condition of its section after sap injection in inoculation holes have been shown in Fig 3. The left side of the figure shows the agarwood samples

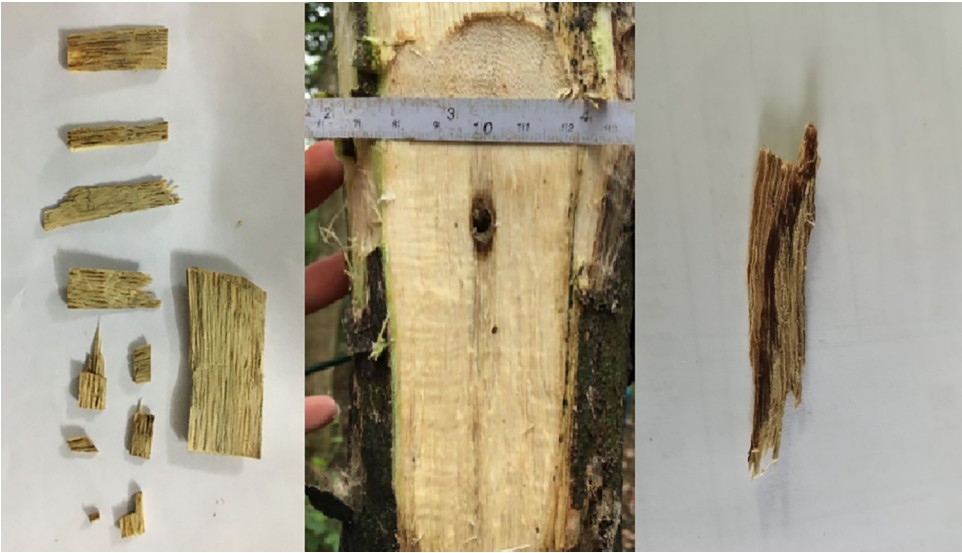

**Fig 3. Agarwood samples.** Note: The left and right pictures show sliced samples and the middle picture shows the growth of the inoculated holes.

generated by the LV, and the right side shows the agarwood samples generated by the H. After removing the bark around the inoculation holes, it can be clearly observed that the color of the xylem around the holes was different from the other parts of the trunk. The discoloration was spread to an area of around 1 cm×2 cm, and the direction of discoloration was mainly longitudinal in upper and lower directions. Samples of discolored wood tissue were found to be free of rotting. Combustion of discolored wood samples generated white scented smoke. Thus, after initial examinations, the discolored xylem was considered agarwood.

The chemical composition of discolored wood samples was analyzed by using GC-MS, and the total ion flow diagram has been shown in Fig 4. The peaks in the total ion flow diagram were analyzed by the MS computer data system and checked against the Nist11 standard MS diagrams. A total of 82 chemical compounds were identified based on manual analysis and identification, as well as by comparing the retention times with findings reported in the previous studies (S3 Table). These 82 chemical compounds included 23 aromatic compounds, 29 sesquiterpenes and 30 other compounds. There were 35 types of same chemical constituents in the 2 groups of incense samples obtained from application of LV and H, respectively. The relative percentages of ether extracts in agarwood samples of LV, H, and CK were 50.22%, 48.71%, and 21.17%, respectively, which were up to 10% as stipulated in the Chinese Pharmacopoeia. The relative proportions of aromatic constituents in agarwood sample of LV, H, and CK were 30.1%, 32.86%, and 7.95%, respectively, while relative contents of selenoids were 10.21%, 11.19%, and 0%, respectively. Moreover, 10 aromatic compounds and 16 sesquiterpenes were common in the agarwood samples of LV and H groups. In LV group, phenol, 2-(phenylmethoxy)- was found to be the major constituent of ether extracts with a mass fraction of 10.56%. On the other hand, 8-Naphthol, 1-(benzyloxy)- content was highest in the ether extracts of H group, with a mass fraction of 7.1%. Both phenol, 2-(phenylmethoxy)- and 8-Naphthol, 1-(benzyloxy)- are aromatic compounds. Sesquiterpenes were present in the form of alcohols and alkenes in the agarwood extracts of LV and H groups. Sesquiterpenes such as

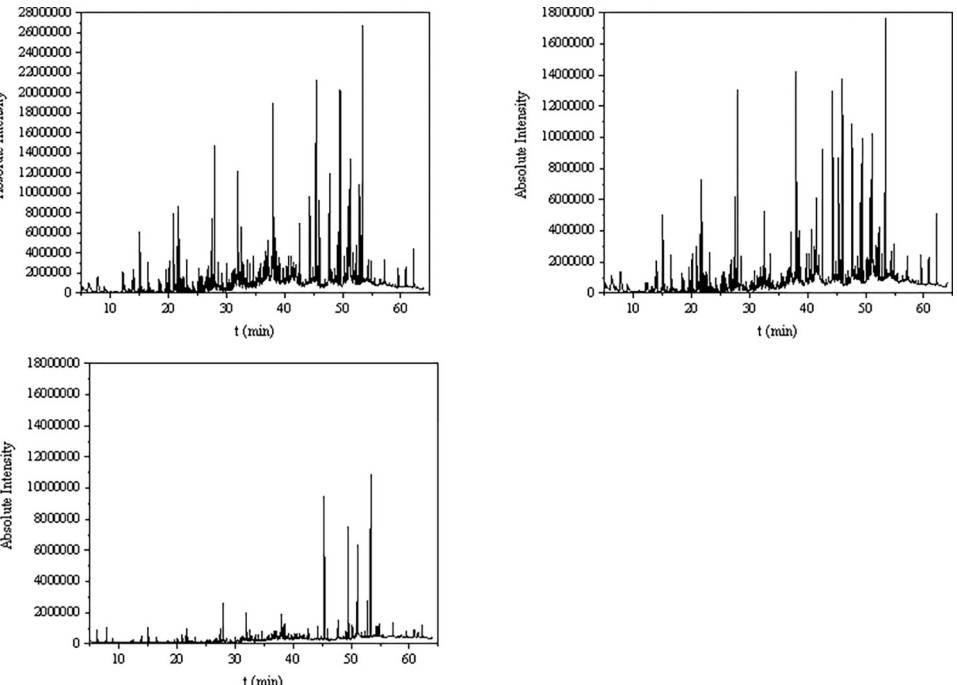

**Fig 4. GC-MS total ion chromatogram of each sample.** Note: From left to right: LV, H, CK.

guaiacolane-type and agarwood spiro-type sesquiterpenes were also detected, with the highest content of β-eudesmol (1.76% and 1.98%, respectively). Agarospirol, a chemical component found only in good quality agarwood, was detected in agarwood extracts of both LV and H groups, with mass fractions of 0.64% and 0.35%, respectively. Total 13 chemical compounds were identified in the samples of CK group, which were devoid of agarwood's main active chemical components: sesquiterpenes and 2-phenylethylchromones. The low sesquiterpene content in the agarwood samples in this study may be due to the loss of low volatile sesquiterpene content during the concentration of ether extract to near dryness, resulting in a low sesquiterpene content. On the other hand, the high aromatic content may be attributed to the fact that the strains were tolerant to benzyl acetone, which is a representative component of the aromatic groups in agarwood. Therefore, the strains might have the ability to synthesize the aromatic components. The absence of chromone components in agarwood samples may be due to insufficient induction time.

## Discussion

In this experiment, *Aquilaria sinensis* trees were subjected to fire and electric shock pre-treatments to promote agarwood formation. The treatment was based on the fact that damaged *Aquilaria sinensis* tree use the phloem for nutrient transport, thus creating the stress to promote agarwood production, without affecting the viability of the tree. On the other hand, electric shock pre-treatment was based on the principle that this treatment changes the ion concentration difference between the interior and exterior parts of plant cell membrane, as well as the distribution of biomolecules; consequently, cell membrane potential changes instantaneously, generating an action potential, which further affect the cell growth and development, and stimulate a self-defense response in the tree to generate agarwood.

The results showed that the dominant fungus in the discolored wood parts was *Fusarium sp*. This finding was in agreement with the previous studies [23–26]. This is the first study reporting the isolation of *Talaromyces sp*. and *Neurospora sp*. from the *Aquilaria sinensis* tree. Previously, Zhang Xiuhuan et al. isolated 42 strains of endophytic fungi from the wood samples of *Aquilaria sinensis*, including 15 strains isolated from the healthy wood and 27 strains isolated from the discolored wood samples. The study reported that the number of fungal species in agarwood-forming parts was much greater than that in healthy tissues of *Aquilaria sinensis* [27]. Wang Lei et al. isolated and identified 50 endophytic fungi from different tissues of *Aquilaria sinensis* trees of different ages. However, the number of fungi obtained from differently aged trees and different tissues varied [28]. Wang et al. obtained 34 fungal isolates from healthy, brown and xylem parts of *Aquilaria crassna* and identified 8 new species, such as *Lasiodiplodia sp*. [29]. Comprehensive review of literature showed that some fungi, such as *Fusarium sp*., *Trichoderma sp*., *Lasiodiplodia sp*. and *Diaporthe sp*. were common among the endophytic fungi isolated from agarwood samples in China and other countries. The variations in the abundance of such fungi and the high degree of specialization in dominant fungi in agarwood may be due to the different microenvironments related to different parts of the plant. For instance, intrinsic factors (such as chemical compositions) and extrinsic factors (such as climate, light, rainfall and soils) may affect the parasitic associations and diversity of endophytic fungi in agarwood.

Due to the complexity of agarwood components including benzyl acetone, some fungi can be intolerant to benzyl acetone but may still show synergistic promoting effect on other agarwood components. In this study, two benzyl acetone-tolerant strains of fungi were obtained after screening experiments. To promote agarwood formation in *Aquilaria sinensis* trees, artificial fungal infusion method was used. This method is based on the principle of dropping the

fungal solution into the trunk of *Aquilaria sinensis*. After injection, the fungal solution is mixed into the trunk sap, and then diffused into other parts of tree. The fungal species introduced in the tree further promote the production of agarwood substances. After 6 months, discolored woody tissues around the inoculation holes were collected. After preliminary assessment, the discolored tissues were observed to possess the apparent properties of agarwood [30].

Sesquiterpenoids, aromatic compounds and (2-phenylethyl) chromones are the main active chemical constituents of agarwood [31]. These are key chemical parameters to assess the quality of agarwood produced after artificial inoculation of fungi in *Aquilaria sinensis* [32]. Quality of agarwood is related to the high relative content of sesquiterpenoids. Yanyan Zhao et al. reported that the volatile oil content in agarwood formed after artificial fungal inoculation was dominated by sesquiterpenes and aromatic constituents [33], and the content of sesquiterpenes was positively related to the induction time [34]. In this study, low sesquiterpene content in the agarwood samples can be attributed to the loss of less volatile sesquiterpene during the near-dry concentration of ether extracts, resulting in a low sesquiterpene content. On the other hand, the high aromatic content may be attributed to the fact that the strains were tolerant to benzyl acetone, which is a representative component of the aromatic groups in agarwood. This suggested that the strains might have the ability to synthesize the aromatic components. The absence of chromone components in agarwood samples may be due to the insufficient induction time. During the process of fungi induced agarwood formation, a symbiotic relationship is formed between the fungal community and the tree. Fungi not only synthesize their own active components similar to those of the host plant, but also have promote the synthesis of active components in the host plant and produce certain secondary metabolites. For example, in *Aquilaria sinensis*, some fungi promote the formation of chromones, while other fungi may promote the formation of aromatic compounds. Therefore, the isolated and purified fungi in this study may have a synergistic promotional role in the formation of other agarwood components [21]. After 6 months of inoculation into *Aquilaria sinensis* trees, both fungi formed a large number of characteristic agarwood substances, such as sesquiterpene compounds and various aromatic compounds. Related studies showed the improved production of agarwood substances from *Aquilaria malaccensis* infected with *Trichoderma sp.*, and the chemical composition of agarwood extracts included: β-eudesmol, aromadendrene oxide, and bulnesol [35]. Similarly, Wang Dongguang et al. used *Trichoderma sp.* to induce agarwood formation in *Aquilaria sinensis* [36], and the chemical composition of generated agarwood extracts was basically the same as the composition of extracts in LV group in this study. *Neurospora sp.* sap was used first time in China to induce the formation of agarwood in *Aquilaria sinensis* trees in this study, and the ether extracts were analyzed by GC-MS. The composition of agarwood substances was not significantly different from those obtained after application of *Melanotus flavolivens* and *Trichoderma sp.* to induce the formation of agarwood in *Aquilaria sinensis* trees [22]. Even though, *Aquilaria sinensis* takes a longer time to produce agarospirol [37], this compound was detected in the all samples in the present study. This indicated that artificial fungal infusion (*Neurospora sp.* and *Trichoderma sp.*) in *Aquilaria sinensis* trees induced agarwood formation within 6 months.

## Conclusions

In this study, *Aquilaria sinensis* trees were pre-treated with fire and electric shock, and the benzylacetone-tolerance of isolated fungi was tested, based on the idea that agarwood-formation promoting fungi should be able to tolerate benzylacetone (one of the main components of agarwood) to a certain extent. Two fungi were found to have the ability to induce the

formation of agarwood. GC-MS analysis revealed that the chemical composition of the agarwood formed by these two fungi was highly similar to the agarwood described in the literature. In just six months, the formed agarwood became unusually rich in terms of aromatic compounds and sesquiterpenes. These findings confirmed the successful development of an efficient and rapid screening method to find fungi that can promote agarwood formation. Overall, this study provided two fungal strains, *Neurospora sp*. and *Trichoderma sp*., with strong ability to induce agarwood formation.

## Supporting information

**S1 Fig. Morphological map of separated and purified parts of fungi.**
(TIF)

**S2 Fig. Gel images of PCR amplification products of H and LV.**
(TIF)

**S3 Fig. Phylogenetic analysis of LV.**
(TIF)

**S4 Fig. Phylogenetic analysis of H.**
(TIF)

**S1 File.**
(TIF)

**S1 Table. Results of preliminary screening test for the concentration of resistance to benzylacetone of each strain.** The size of the bacterial circle diameter is the average of 3 repeated experiments; "*" in the same industry indicates that the difference is significant at the $P<0.05$ level, and "**" indicates that the difference is extremely significant at the $P<0.01$ level.
(TIF)

**S2 Table. Results of re-screening test for the concentration of resistance to benzylacetone of each strain.** The size of the bacterial circle diameter is the average of 3 repeated experiments; "*" in the same industry indicates that the difference is significant at the $P<0.05$ level, and "**" indicates that the difference is extremely significant at the $P<0.01$ level.
(TIF)

**S3 Table. Chemical components in ether extracts of 3 samples.** Mean's not detected.
(TIF)

## Author Contributions

**Conceptualization:** Guoying Zhou.

**Supervision:** Guoying Zhou, Junang Liu.

**Writing – original draft:** Chuang Liu.

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
