## [Decision Letter · Decision Letter 0]

20 Oct 2023

PONE-D-23-28830Induction of agarwood formation by artificial inoculation with fungi: Isolation and screening of fungi promoting agarwood formation in Aquilaria sinensis treesPLOS ONE

Dear Dr. chuang,

Thank you for submitting your manuscript to PLOS ONE. After careful consideration, we feel that it has merit but does not fully meet PLOS ONE’s publication criteria as it currently stands. Therefore, we invite you to submit a revised version of the manuscript that addresses the points raised during the review process.

We look forward to receiving your revised manuscript.

Kind regards,

Niraj Agarwala, Ph.D.

Academic Editor

PLOS ONE

Journal Requirements:

5. Thank you for stating in your Funding Statement: 

Supported by the Sub-projects of the 13th Five-Year National Key R&D Plan (2016YFD0600601-1), and Technological Innovation Special Project of Research Institutes of Hainan Province (jscx202028).

Reviewers' comments:

Reviewer's Responses to Questions

**Comments to the Author**

1. Is the manuscript technically sound, and do the data support the conclusions?

Reviewer #1: Partly

Reviewer #2: Partly

2. Has the statistical analysis been performed appropriately and rigorously? 

Reviewer #1: No

Reviewer #2: N/A

3. Have the authors made all data underlying the findings in their manuscript fully available?

Reviewer #1: Yes

Reviewer #2: Yes

4. Is the manuscript presented in an intelligible fashion and written in standard English?

Reviewer #1: No

Reviewer #2: Yes

5. Review Comments to the Author

Reviewer #1: Overall Comment

The publication doesn’t meet the standards of the PLOSone journal. The manuscript needs major refinement at usage of English, Data and its appropriate presentation and scientific elaboration and novelty of the study needs to be highlighted upon in the manuscript.

1. Very poorly written manuscript, major revision at writing level needed. Statements made are not complete, very undecipherable, data is basic going by standards of PLOSone, presentation needs major changes to make it suitable for publication. Methodology part is only lucid and elaborately written, all other parts need major revision.

2. There have been many publications on artificial inoculation in A sinensis recently (Ma et al., 2021, Jalil et al 2021 etc.), what is the innovative method used in the study needs to be highlighted.

3. The comparative analysis of the data in form of a discussion on the new knowledge gained in background all the existing knowledge is totally lacking. That makes the manuscript sound vague though the data has the potential to make a good read. No scientific reasoning for the experimentation could be found in the text. The conclusion is confusing, what is being concluded is not clear at all.

4. Background and sound scientific basis for carrying out the experimentation is not reflected by current version of the manuscript which needs to be majorly addressed upon.

Specific comments

For Abstract

1. Overall issues with English hugely noted, like the statement in abstract “ In this study, different aroma-making treatments were applied to Aquilaria sinensis plantation pure forests to isolate and purify the dominant fungi on the tree surface and discolored wood, and the strains were identified by molecular biology.

2. Why were stains screened for resistance using benzylacetone? Explanation needed.

3. “The Trichoderma sp. and Neurospora sp. samples yielded ethyl ether extracts of 3.62% and 4.04%, respectively” what significance does this statement has? Statement that make characteristic contribution to the results should to be highlighted in abstract.

4. The sum of the relative contents of the aromatic components was 30.1% and 32.86%, respectively. What significance does this statement has? Statement that make characteristic contribution to the results need to be highlighted in abstract.

5.The relative contents of sesquiterpenes were 10.21% and 11.19%. Trichoderma sp., Neurospora sp. were able to effectively induce the production of agarwood within 6 months, and their good effect on agarwood formation and shortening the time of agarwood formation. Not clear what new information/ knowledge or method has been created by this study.

6. What does the statement “It provides effective support for the expansion of the fungal library that promotes agarwood formation” substantialize or conclude? Choice of word usages have diluted the real meaning of the statement.

From main text

7.What is Introduce? Should be Introduction.

8.Its trunk does not produce agarwood if there is no external damage, only after natural or man-made injuries such as slashes, injuries and diseases, insects and ants, lightning strikes, etc….is a incomplete sentence

9.Agarwood is mildly warm in nature. How??

10.Since when Agarwood has been used as Precious spice? Please quote references.

11.Historically, China's agarwood resources are very rich, with the discovery of the medicinal value of agarwood ……………………….. field of Aquilaria sinensis has been very little, was sporadic distribution. Is a very complex statement and doesn’t make a good read. This needs simplification and usage of proper English to convey the core thought.

12. There have been many publication on artificial inoculation in A sinensis recently, a comparative account of the same would have been more interesting read which could have highlighted upon the advantages of the method devised in the study.

13.What is stained wood?

14.Why Benzylacetone was selected as a resistance indicator needs to be elaborated? What added advantage it confers to the study is not apparent.

15.Justification for carrying out the study in not adequate in view of the available already.

16.The method used is amalgamation of too many method Physical, heat, electric current creating a state of over stress to the plant , where within under humid condition any fungal pathogen attack on the bark is possible. The fungus growing may not necessarily be endofungal community specific to the plant. A statement or clarification in this regard would be welcome by readers.

17.The prepared sample was weighed 3 g wet weight into a 50 mL centrifuge tube and 15 mL of ether was added to extract the sample ………………………………….to obtain the ether extract, the extraction was repeated three times, then the extracts were combined and weighed in a fume hood after drying and the volatile oil content was calculated. How was it calculated? Statement also needs restructuring to make it easy to understand and read.

18.Table 1 showing that benzylacetone produced varying degrees of inhibition on all 26 isolated fungal strains can be a shown as supplementary data.

19.What is LV, H and CK need to be introduced to readers more properly in a one statement.

20.The discoloured wood was subjected to chemical compositional identification and the yield of ethyl ether extract was 3.62%, 4.04% and 1.94% for LV, H and CK respectively. Its GC-MS total ion flow (Figure 5). What does this statement mean?

21.“signifying the relative stability of the sesquiterpenes” How? Elaborate.

22.Which lacked any of the 3 major active chemical components of Agarwood. What are these major active chemicals name them here.

23.The comparative analysis of the data in form of a discussion on the new knowledge gained in background all the existing knowledge is totally lacking.

24.Discussion of results is missing. That makes the manuscript sound vague. There is no reference to previous work in same field. There is basic data but very poorly presented. No scientific reasoning for the experimentation could be found in the text.

25.Table 1 and 2 should be made supplementary data.

26.Figure 3 should be made supplementary data.

27.Figure legends for Figure 4 and 5 poorly presented. Figure legend for 4 is missing.

28.The conclusion is confusing. What is the conclusion of the study?

29.“The fungi selected in this paper have good effects in promoting Agarwood production and shortening Agarwood formation time in Aquilaria sinensis trees” How? Prove the statement by comparing with previous available data needed in text.

30. Fusarium and Trichoderma are popular choices for inoculation, the study should list out the novelty of the present work more clearly.

31. Referencing for study not adequate.

Reviewer #2: Title: Induction of agarwood formation by artificial inoculation with fungi: Isolation and screening of fungi promoting agarwood formation in Aquilaria sinensis trees

Summary: In order to efficiently and accurately screen Aquilaria sinensis aroma-promoting fungi, expand the library of Aquilaria sinensis aroma-promoting fungi, and shorten the aroma-forming cycle of Aquilaria sinensis. Present results provide effective support for the expansion of the fungal library that promotes agarwood formation. The authors attempted a good approach to perform the following study. Following are my suggestions for further improvements.

Review comments:

1. Abstract need to be more summarized with the actual study results.

2. The introduction section needs to be adhered to with the proposed title. Also needs to be referred to relevant published literature. The following published articles might be helpful in the enhancements of the introduction section.

- Production of volatile compounds by a variety of fungi in artificially inoculated and naturally infected Aquilaria malaccensis. Current Microbiology, 79(5), 151.

- An is gene-mediated molecular detection of fungi associated with natural and artificial agarwood from Aquilaria malaccensis. Journal of microbiology, biotechnology, and food sciences, e9465-e9465.

- Fungi mediated agarwood (A. malaccensis) production and their pharmaceutical applications: A systematic review. International Journal of Plant-Based Pharmaceuticals, 2(2), 261-270.

3. The novelty of the work needs to be highlighted. The objectives of the work can be elaborated for reproduction of the work.

4. Methods need to be reproducible. The authors provided too descriptive methods. I suggest shortening them and extra parts can be moved to the results section.

5. Also, the methods can be separated into sub-topics for ease of access. Like GCMS analysis and the different shock treatments.

6. For the GCMS analysis following published literature might be useful such as Gas chromatography analysis of the microwave-aided extracted agarwood oil from physically induced Aquilaria malaccensis trees in Northern Thailand. Maejo International Journal of Energy and Environmental Communication, 4(3), 52-55.

7. In the methods, the authors mentioned the PCR amplification, but in the results, I am unable to find the results related to it. No gel images or DNA analysis results were presented.

8. If the authors plan to include Table 3 can be moved to supplementary information. AS Figure 5 shows the chromatographs.

9. The results need to be enhanced with the help of literature. The discussion section is not comprehensive enough to refer to published articles for improvements.

10. The grammatical errors and the English language need to be refined.

6. PLOS authors have the option to publish the peer review history of their article (what does this mean?). If published, this will include your full peer review and any attached files.

Reviewer #1: **Yes: **Sofia Banu

Reviewer #2: No

---

## [Author Response · Author response to Decision Letter 0]

8 Nov 2023

Response to Reviewers

According to the review comments from 2 editors, this paper has an overall problem with the English language, so I made a complete revision of my manuscript to meet the journal requirements.

Reviewer #1

1、Why were stains screened for resistance using benzylacetone? 

Studies on endophytic fungi growing on medicinal plants have shown that endophytic fungi can not only synthesize their own active components, but also have the ability to promote the synthesis of active components by the host plant . Fungi promoting the formation of agarwood are firstly tolerant to some extent to the active ingredients of agarwood. The composition of agarwood is complex, consisting of three main components: sesquiterpenes, 2-phenylethylchromones, and aromatic compounds. Benzyl acetone is a representative compound of the aromatic content in agarwood .

2、“The Trichoderma sp. and Neurospora sp. samples yielded ethyl ether extracts of 3.62% and 4.04%, respectively” what significance does this statement has? 

“The Trichoderma sp. and Neurospora sp. samples yielded ethyl ether extracts of 3.62% and 4.04%, respectively” In Chinese literature, most of them will write this "extraction rate" to express the accuracy of the extraction method. This has been modified in the text.

3、The sum of the relative contents of the aromatic components was 30.1% and 32.86%, respectively. What significance does this statement has? Statement that make characteristic contribution to the results need to be highlighted in abstract.

These two agarwood-specific chemical constituents accounted for a large proportion of the total chemical composition, which showed that the generated agarwood was of good quality.

4、The relative contents of sesquiterpenes were 10.21% and 11.19%. Trichoderma sp., Neurospora sp. were able to effectively induce the production of agarwood within 6 months, and their good effect on agarwood formation and shortening the time of agarwood formation. Not clear what new information/ knowledge or method has been created by this study.

In this study, pretreatment was carried out before exposing the Aquilaria sinensis trees to fungi. Moreover, isolated fungi were screened for resistance against benzyl acetone, which is a major component of agarwood. Artificial inoculation of fungi in Aquilaria sinensis trees was able to effectively induce the formation of agarwood over a 6-month period, The produced agarwood incense were analyzed by gas chromatography-mass spectroscopy (GC-MS). The method presented here is able to narrow down the range of fungi that promote agarwood formation, and reduce the time and monetary cost of screening these fungi. The study would expand the library of fungi known for promoting agarwood formation.

5、What does the statement “It provides effective support for the expansion of the fungal library that promotes agarwood formation” substantialize or conclude? Choice of word usages have diluted the real meaning of the statement.

Changes have been made in the text

6、What is Introduce? Should be Introduction.

Changes have been made in the text

7、Its trunk does not produce agarwood if there is no external damage, only after natural or man-made injuries such as slashes, injuries and diseases, insects and ants, lightning strikes, etc….is a incomplete sentence

Changes have been made in the text

8、Agarwood is mildly warm in nature. How??

Changes have been made in the text. In Chinese medicine, agarwood is mild in nature.

9、Since when Agarwood has been used as Precious spice? Please quote references.

Changes have been made and references have been cited

10、Historically, China's agarwood resources are very rich, with the discovery of the medicinal value of agarwood ……………………….. field of Aquilaria sinensis has been very little, was sporadic distribution. Is a very complex statement and doesn’t make a good read. This needs simplification and usage of proper English to convey the core thought.

Changes have been made in the text

11、11.There have been many publication on artificial inoculation in A sinensis recently, a comparative account of the same would have been more interesting read which could have highlighted upon the advantages of the method devised in the study.

Changes have been made in the text

12.What is stained wood?

Poorly worded, should be wood that changes colour

13.Why Benzylacetone was selected as a resistance indicator needs to be elaborated? What added advantage it confers to the study is not apparent.

Changes have been made in the text

14.Justification for carrying out the study in not adequate in view of the available already.

Changes have been made in the text

15.The method used is amalgamation of too many method Physical, heat, electric current creating a state of over stress to the plant , where within under humid condition any fungal pathogen attack on the bark is possible. The fungus growing may not necessarily be endofungal community specific to the plant. A statement or clarification in this regard would be welcome by readers.

At first, we were trying to take endophytic fungi after fire-burning Aquilaria sinensis trees, but it was interesting to observe that the epidermis of Aquilaria sinensis trees was covered with fungi, and then we wanted to start the experiment with that colony. It is not said that the fungi covering the epidermis of Aquilaria sinensis trees are endophytic fungi.

16.The prepared sample was weighed 3 g wet weight into a 50 mL centrifuge tube and 15 mL of ether was added to extract the sample ………………………………….to obtain the ether extract, the extraction was repeated three times, then the extracts were combined and weighed in a fume hood after drying and the volatile oil content was calculated. How was it calculated? Statement also needs restructuring to make it easy to understand and read.

Changes have been made in the text

17.Table 1 showing that benzylacetone produced varying degrees of inhibition on all 26 isolated fungal strains can be a shown as supplementary data.

Changes have been made in the text

18.What is LV, H and CK need to be introduced to readers more properly in a one statement.

Changes have been made in the text

19.The discoloured wood was subjected to chemical compositional identification and the yield of ethyl ether extract was 3.62%, 4.04% and 1.94% for LV, H and CK respectively. Its GC-MS total ion flow (Figure 5). What does this statement mean?

Changes have been made in the text

20.“signifying the relative stability of the sesquiterpenes” How? Elaborate.

This paper would like to express that in this study sesquiterpenoids are more stable as compared to aromatic compounds.

21.Which lacked any of the 3 major active chemical components of Agarwood. What are these major active chemicals name them here.

Changes have been made in the text

22.The comparative analysis of the data in form of a discussion on the new knowledge gained in background all the existing knowledge is totally lacking.

Changes have been made in the text

23.Discussion of results is missing. That makes the manuscript sound vague. There is no reference to previous work in same field. There is basic data but very poorly presented. No scientific reasoning for the experimentation could be found in the text.

Changes have been made in the text

24.Table 1 and 2 should be made supplementary data.

Changes have been made in the text

25.Figure 3 should be made supplementary data.

Changes have been made in the text

26.Figure legends for Figure 4 and 5 poorly presented. Figure legend for 4 is missing.

Changes have been made in the text

27.The conclusion is confusing. What is the conclusion of the study?

Changes have been made in the text

28.“The fungi selected in this paper have good effects in promoting Agarwood production and shortening Agarwood formation time in Aquilaria sinensis trees” How? Prove the statement by comparing with previous available data needed in text.

Changes have been made in the text

29. Fusarium and Trichoderma are popular choices for inoculation, the study should list out the novelty of the present work more clearly.

Neurospora sp. sap was used first time in China to induce the formation of agarwood in Aquilaria sinensis trees in this study. In this study, pretreatment was carried out before exposing the Aquilaria sinensis trees to fungi. Moreover, isolated fungi were screened for resistance against benzyl acetone. The method presented here is able to narrow down the range of fungi that promote agarwood formation, and reduce the time and monetary cost of screening these fungi. The study would expand the library of fungi known for promoting agarwood formation.

30. Referencing for study not adequate.

Changes have been made in the text

Reviewer #2

1.Abstract need to be more summarized with the actual study results.

Changes have been made in the text

2. The introduction section needs to be adhered to with the proposed title. Also needs to be referred to relevant published literature. The following published articles might be helpful in the enhancements of the introduction section.

Changes have been made in the text

3. The novelty of the work needs to be highlighted. The objectives of the work can be elaborated for reproduction of the work.

Changes have been made in the text

4. Methods need to be reproducible. The authors provided too descriptive methods. I suggest shortening them and extra parts can be moved to the results section.

Changes have been made in the text

5. Also, the methods can be separated into sub-topics for ease of access. Like GCMS analysis and the different shock treatments.

Changes have been made in the text

6. For the GCMS analysis following published literature might be useful such as Gas chromatography analysis of the microwave-aided extracted agarwood oil from physically induced Aquilaria malaccensis trees in Northern Thailand. Maejo International Journal of Energy and Environmental Communication, 4(3), 52-55.

Changes have been made in the text

7. In the methods, the authors mentioned the PCR amplification, but in the results, I am unable to find the results related to it. No gel images or DNA analysis results were presented.

Changes have been made in the text.

8. If the authors plan to include Table 3 can be moved to supplementary information. AS Figure 5 shows the chromatographs.

Changes have been made in the text.

9. The results need to be enhanced with the help of literature. The discussion section is not comprehensive enough to refer to published articles for improvements.

Changes have been made in the text.

10. The grammatical errors and the English language need to be refined.

Changes have been made in the text.

---

## [Decision Letter · Decision Letter 1]

18 Dec 2023

PONE-D-23-28830R1真菌的分离和筛选以增强沉香树的形成PLOS ONE

Dear Dr. chuang,

Thank you for submitting your manuscript to PLOS ONE. After careful consideration, we feel that it has merit but does not fully meet PLOS ONE’s publication criteria as it currently stands. Therefore, we invite you to submit a revised version of the manuscript that addresses the points raised during the review process. 

We look forward to receiving your revised manuscript.

Kind regards,

Niraj Agarwala, Ph.D.

Academic Editor

PLOS ONE

Reviewers' comments:

Reviewer's Responses to Questions

**Comments to the Author**

1. If the authors have adequately addressed your comments raised in a previous round of review and you feel that this manuscript is now acceptable for publication, you may indicate that here to bypass the “Comments to the Author” section, enter your conflict of interest statement in the “Confidential to Editor” section, and submit your "Accept" recommendation.

Reviewer #1: All comments have been addressed

Reviewer #2: All comments have been addressed

2. Is the manuscript technically sound, and do the data support the conclusions?

Reviewer #1: Partly

Reviewer #2: Yes

3. Has the statistical analysis been performed appropriately and rigorously? 

Reviewer #1: No

Reviewer #2: N/A

4. Have the authors made all data underlying the findings in their manuscript fully available?

Reviewer #1: No

Reviewer #2: Yes

5. Is the manuscript presented in an intelligible fashion and written in standard English?

Reviewer #1: No

Reviewer #2: No

6. Review Comments to the Author

Reviewer #1: Overall Comment:

Though the Authors took the effort to modify the manuscript. Still there is a huge scope to modify it further to make it a much appreciable piece of work. A major revision is still needed to make it a scientifically presented manuscript.

From the previous comments 3 comments where not answered adequately

1. A thorough English check of the manuscript

2. Comments 15 Not adequately answered. A more insightful explanation needed for the amalgamation of too many methods (Physical, heat, electric current creating a state of over stress to the plant). Justification on the role of the pretreatment needed to be elaborated more scientifically in discussion

3. Comment 19: Not adequately answered.

Abstract:

1. Agarwood is formed when Aquilaria sinensis trees are harmed naturally or by humans: Modify the statement.

2. In this study, dominant fungal species were isolated and purified from the surfaces and electroshock holes of Aquilaria sinensis trees by fire and electroshock treatments: Modify the statement

3. In agarwood products formed by using Trichoderma sp. And Neurospora sp., the proportions of ether extracts were 50.22% and 48.71%, respectively, exceeding the 10% threshold specified by the Chinese Pharmacopoeia: Modify to make it applicable to International Readers too. What is agarwood products? Its just raesin deposition.

4. Fungal application shortened the time of natural agarwood formation: If you have applied fungal inoculum how is its natural infection, its induced formation. Modify the statement.

Introduction:

5. Fungi promoting the formation of agarwood are firstly tolerant to some extent to the active ingredients of agarwood: Quote the reference for the statement.

6. In this study, pretreatment was carried out before exposing the Aquilaria sinensis trees to fungi. Moreover, isolated fungi were screened for resistance…………………… The study would expand the library of fungi known for promoting agarwood formation: These are concluding lines used by author in conclusion also. These cannot be part of Introduction segment.

7. The method presented here is able to narrow down the range of fungi that promote agarwood formation, and reduce the time and monetary cost of screening these fungi. The study would expand the library of fungi known for promoting agarwood formation: Two contradictory statements made in same para, Modify.

Materials and methods

8. Aquilaria sinensis needs to be italic

9. The discolored wood of Aquilaria sinensis was collected after treatment by the electric shock pretreatment for 3 months

10. 1.5ml should be 1.5 ml. Follow international norms Value space SI Units.

11. -20℃ should be -20 ℃.

12. 5cm should be 5 cm. Follow this throughout the manuscript.

13. Fig.2 Map of the dominant colony on the surface of the tree and the dripping and sampling position of the bacterial liquid: Should be Fig.2. Picture of the of the dominant colony on the surface of the tree, the dripping and sampling position of the fungal liquid.

14. Evidences in form of Photographs are not adequate, some more detailing needed, e.g. pictures of inoculation with GI tag pictures etc.

Results

1. Using molecular identification, the dominant fungal strains on the surface of Aquilaria sinensis were initially identified as Trichoderma sp. and Neurospora sp., which were labelled as LV and H, respectively: Please provide the evidences of identification eg. Amplification of the PCR products (gel images), Phylogenetic analysis etc as supplementary data

2. The red and green fungal colonies that covered the burnt parts of the trees after the fire treatment (Fig 2) were attributed to the decay and removal of other microorganisms from the trees due to fire, resulting in the vigorous growth of Trichoderma sp. and Neurospora sp.: Is a very very vague statement. Either remove it or provide evidences for stating the same.

3. The sentences “The red and green fungal colonies that covered the burnt parts of the trees after the fire treatment………………………………. intrinsic factors (such as chemical compositions) and extrinsic factors (such as the climate, light, rainfall and soils) may affect the parasitism and diversity of endophytic fungi in agarwood” are not results need to be moved to discussion section.

4. In the sentence “Promotion of agarwood formation in Aquilaria sinensis by artificial inoculation of benzyl acetone-tolerant fungi” Aquilaria sinensis needs to be in italics.

5. The sentences “The low sesquiterpene content in the agarwood……………………………………absence of chromone components in agarwood samples may be due to insufficient induction time” are explanations to the results obtained and would be appropriate if are moved to discussion section.

6. Figure legends are poorly presented

Discussion:

1. After injection, the fungal solution is mixed into the trunk sap, and then diffused into other parts of tree: Is the statement correct?

Conclusion:

1. Needs further modification to highlight upon the novelty of the work. The continuity between the statements is missing, which the authors need to improve upon majorly. The conclusion needs to be rewritten.

Reviewer #2: The authors successfully responded to the comments raised. The revised version is acceptable for publication.

7. PLOS authors have the option to publish the peer review history of their article (what does this mean?). If published, this will include your full peer review and any attached files.

Reviewer #1: No

Reviewer #2: No

---

## [Author Response · Author response to Decision Letter 1]

25 Dec 2023

Response to Reviewers

According to the review comments from editors, this paper has an overall problem with the English language, so I made a complete revision of my manuscript to meet the journal requirements.

From the previous comments 3 comments where not answered adequately

1.A thorough English check of the manuscript 

The manuscript has been thoroughly reviewed in English and revised.

2.Comments 15 Not adequately answered. A more insightful explanation needed for the amalgamation of too many methods (Physical, heat, electric current creating a state of over stress to the plant). Justification on the role of the pretreatment needed to be elaborated more scientifically in discussion

I have added the understanding of the pretreatment effect in the discussion.

3.Comment 19: Not adequately answered. 

It has been changed in the new manuscript, previously meaning that after ether extraction, the material finally extracted the yield of the essential oil.

Abstract:

1.Agarwood is formed when Aquilaria sinensis trees are harmed naturally or by humans: Modify the statement.

The statement has been revised

2.In this study, dominant fungal species were isolated and purified from the surfaces and electroshock holes of Aquilaria sinensis trees by fire and electroshock treatments: Modify the statement

The statement has been revised

3.In agarwood products formed by using Trichoderma sp. And Neurospora sp., the proportions of ether extracts were 50.22% and 48.71%, respectively, exceeding the 10% threshold specified by the Chinese Pharmacopoeia: Modify to make it applicable to International Readers too. What is agarwood products? Its just raesin deposition.

The statement has been revised.In this paper, the agarwood product means that the discolored wood obtained on the tree through the action of fungi is identified as containing agarwood ingredients after the chemical composition. The wood is the agarwood product.

4.Fungal application shortened the time of natural agarwood formation: If you have applied fungal inoculum how is its natural infection, its induced formation. Modify the statement.

The statement has been revised

Introduction:

5.Fungi promoting the formation of agarwood are firstly tolerant to some extent to the active ingredients of agarwood: Quote the reference for the statement.

Have already cited reference

6.In this study, pretreatment was carried out before exposing the Aquilaria sinensis trees to fungi. Moreover, isolated fungi were screened for resistance…………………… The study would expand the library of fungi known for promoting agarwood formation: These are concluding lines used by author in conclusion also. These cannot be part of Introduction segment. 

Have deleted

7.The method presented here is able to narrow down the range of fungi that promote agarwood formation, and reduce the time and monetary cost of screening these fungi. The study would expand the library of fungi known for promoting agarwood formation: Two contradictory statements made in same para, Modify.

Has been modified

Materials and methods

8.Aquilaria sinensis needs to be italic

Has been modified

9.The discolored wood of Aquilaria sinensis was collected after treatment by the electric shock pretreatment for 3 months

Has been modified

10.1.5ml should be 1.5 ml. Follow international norms Value space SI Units.

Has been modified

11.-20℃ should be -20 ℃.

Has been modified

12.5cm should be 5 cm. Follow this throughout the manuscript.

Has been modified

13.Fig.2 Map of the dominant colony on the surface of the tree and the dripping and sampling position of the bacterial liquid: Should be Fig.2. Picture of the of the dominant colony on the surface of the tree, the dripping and sampling position of the fungal liquid.

Has been modified

14.Evidences in form of Photographs are not adequate, some more detailing needed, e.g. pictures of inoculation with GI tag pictures etc.

More detailed information has been added to the attachment

Results

1.Using molecular identification, the dominant fungal strains on the surface of Aquilaria sinensis were initially identified as Trichoderma sp. and Neurospora sp., which were labelled as LV and H, respectively: Please provide the evidences of identification eg. Amplification of the PCR products (gel images), Phylogenetic analysis etc as supplementary data

The alignment information of the sequencing results has been provided in the attachment

2.The red and green fungal colonies that covered the burnt parts of the trees after the fire treatment (Fig 2) were attributed to the decay and removal of other microorganisms from the trees due to fire, resulting in the vigorous growth of Trichoderma sp. and Neurospora sp.: Is a very very vague statement. Either remove it or provide evidences for stating the same. 

Has been modified

3.The sentences “The red and green fungal colonies that covered the burnt parts of the trees after the fire treatment………………………………. intrinsic factors (such as chemical compositions) and extrinsic factors (such as the climate, light, rainfall and soils) may affect the parasitism and diversity of endophytic fungi in agarwood” are not results need to be moved to discussion section.

Has been modified

4.In the sentence “Promotion of agarwood formation in Aquilaria sinensis by artificial inoculation of benzyl acetone-tolerant fungi” Aquilaria sinensis needs to be in italics.

Has been modified

5.The sentences “The low sesquiterpene content in the agarwood……………………………………absence of chromone components in agarwood samples may be due to insufficient induction time” are explanations to the results obtained and would be appropriate if are moved to discussion section.

Has been modified

6.Figure legends are poorly presented

Has been modified,But I don't know how to modify it even better

Discussion:

1.After injection, the fungal solution is mixed into the trunk sap, and then diffused into other parts of tree: Is the statement correct?

I think correctly, I inject the fluid into the trunk and definitely spread to the rest of the tree. Maybe the process is not very accurate, but the result should be what I said.

Conclusion:

1.Needs further modification to highlight upon the novelty of the work. The continuity between the statements is missing, which the authors need to improve upon majorly. The conclusion needs to be rewritten. 

Has been modified

Finally, I wish you a merry Christmas, if there are any deficiencies, please understand, I wish you a happy life, everything goes well.

---

## [Decision Letter · Decision Letter 2]

26 Mar 2024

PONE-D-23-28830R2Isolation and screening of fungi for enhanced agarwood formation in Aquilaria sinensis treesPLOS ONE

Dear Dr. chuang,

Thank you for submitting your manuscript to PLOS ONE. After careful consideration, we feel that it has merit but does not fully meet PLOS ONE’s publication criteria as it currently stands. Therefore, we invite you to submit a revised version of the manuscript that addresses the points raised during the review process.

**Sorry for the delay in getting the reviewers report regarding the manuscript. The reviewers recommend some minor revisions to the manuscript for acceptance. Please address the reviewers comment.**

We look forward to receiving your revised manuscript.

Kind regards,

Niraj Agarwala, Ph.D.

Academic Editor

PLOS ONE

Journal Requirements:

Reviewers' comments:

Reviewer's Responses to Questions

**Comments to the Author**

1. If the authors have adequately addressed your comments raised in a previous round of review and you feel that this manuscript is now acceptable for publication, you may indicate that here to bypass the “Comments to the Author” section, enter your conflict of interest statement in the “Confidential to Editor” section, and submit your "Accept" recommendation.

Reviewer #1: All comments have been addressed

2. Is the manuscript technically sound, and do the data support the conclusions?

Reviewer #1: (No Response)

3. Has the statistical analysis been performed appropriately and rigorously? 

Reviewer #1: Yes

4. Have the authors made all data underlying the findings in their manuscript fully available?

Reviewer #1: Yes

5. Is the manuscript presented in an intelligible fashion and written in standard English?

Reviewer #1: Yes

6. Review Comments to the Author

Reviewer #1: Abstract:

1. Agarwood is not produced naturally; it only forms after natural or man-made damage to Aquilaria sinensis. Replace man-made damage with suitable wording

Introduction:

1. The naturally occurring agarwood has long outstripped the market demand, and at the end of the..Replace outstripped with outrun

2. Benzyl acetone is a representative compound of the aromatic content in agarwood……….the statement needs to be in context. Here it is just a hanging statement.

Results

1. Screening of benzyl acetone-tolerant fungi : a certain extent, the isolated and purified fungal strains were screened for tolerance to benzyl acetone.

2. GC-MS detection of agarwood samples ……where is th evidence for the same provided.

3. The agarwood products…..use statement “The agarwood”

4. Fig.3 Agarwood samples…………Reasin deposition seen in figure is very very less than claimed in the text. Please explain.

5. For the query Using molecular identification, the dominant fungal strains on the surface of Aquilaria sinensis were initially identified as Trichoderma sp. and Neurospora sp., which were labelled as LV and H, respectively: Please provide the evidences of identification eg. Amplification of the PCR products (gel images), Phylogenetic analysis etc as supplementary data

6. The alignment information of the sequencing results has been provided which I think is not adequate, please provide the gel images etc.

7. PLOS authors have the option to publish the peer review history of their article (what does this mean?). If published, this will include your full peer review and any attached files.

Reviewer #1: **Yes: **Sofia Banu

---

## [Author Response · Author response to Decision Letter 2]

25 Apr 2024

Response to Reviewers

Abstract:

1.Agarwood is not produced naturally; it only forms after natural or man-made damage to Aquilaria sinensis. Replace man-made damage with suitable wording

Modifications have been made

Introduction: 

1.The naturally occurring agarwood has long outstripped the market demand, and at the end of the..Replace outstripped with outrun

Modifications have been made

2.Benzyl acetone is a representative compound of the aromatic content in agarwood……….the statement needs to be in context. Here it is just a hanging statement.

Modifications have been made

Results

1.Screening of benzyl acetone-tolerant fungi : a certain extent, the isolated and purified fungal strains were screened for tolerance to benzyl acetone.

Modifications have been made

2.GC-MS detection of agarwood samples ……where is th evidence for the same provided.

Original documentation of the GC-MS test has been provided.

3.The agarwood products…..use statement “The agarwood”

Modifications have been made

4.Fig.3 Agarwood samples…………Reasin deposition seen in figure is very very less than claimed in the text. Please explain.

The image has been replaced.As can be seen in the supplementary document (Schematic representation of the fungal inoculation with Aquilaria sinensis)

5.For the query Using molecular identification, the dominant fungal strains on the surface of Aquilaria sinensis were initially identified as Trichoderma sp. and Neurospora sp., which were labelled as LV and H, respectively: Please provide the evidences of identification eg. Amplification of the PCR products (gel images), Phylogenetic analysis etc as supplementary data 

Gel images and phylogenetic trees of PCR amplification products of H and LV have been provided in the Supplementary Material

6.The alignment information of the sequencing results has been provided which I think is not adequate, please provide the gel images etc.

Gel images and phylogenetic trees of PCR amplification products of H and LV have been provided in the Supplementary Material

---

## [Editor Report · Decision Letter 3]

7 May 2024

PONE-D-23-28830R3Isolation and screening of fungi for enhanced agarwood formation in Aquilaria sinensis treesPLOS ONE

Dear Dr. chuang,

Thank you for submitting your manuscript to PLOS ONE. After careful consideration, we feel that it has merit but does not fully meet PLOS ONE’s publication criteria as it currently stands. Therefore, we invite you to submit a revised version of the manuscript that addresses the points raised during the review process.

We look forward to receiving your revised manuscript.

Kind regards,

Niraj Agarwala, Ph.D.

Academic Editor

PLOS ONE

Journal Requirements:

Additional Editor Comments:

Dear authors,

I can see that most of the queries raised by the reviewers are now resolved. In my view the article is almost ready for publication, some minor modifications will enhance the readability and scientific merit of the article.

1. Figure 2 and Figure 3 descriptions can be more specific. For figure 2, specifically mention which inoculation method results in agarwood formation shown in the photograph. For figure.3 specifically mention the type of sample used for GC MS analysis for each. Not in the form of Note. If abbreviations are used then also provide full form in figure captions.

2. Regarding methodology used for GC-MS analysis reference 20 is cited, for which it is mentioned that full text is only available in Chinese. Give complete details of GC-MS methods used in the study in the manuscript

I hope my inputs will be useful to improve your manuscript. Best wishes

Regards

Niraj

---

## [Author Response · Author response to Decision Letter 3]

8 May 2024

1.Figure 2 and Figure 3 descriptions can be more specific. For figure 2, specifically mention which inoculation method results in agarwood formation shown in the photograph. For figure.3 specifically mention the type of sample used for GC MS analysis for each. Not in the form of Note. If abbreviations are used then also provide full form in figure captions.

The left side of Fig. 2 shows the dominant fungi in the tree after fire treatment. After isolating and purifying the fungi, the fungal solution was injected into the tree using the method shown in the right side of the figure, so it is the fungal injection method shown in the right side of the figure that induces the formation of agarwood. This is explained in "Artificial inoculation of fungi to induce the formation of agarwood in Aquilaria sinensis".

For Figure 3, the type of each sample used for GC MS analysis has been described.

2. Regarding methodology used for GC-MS analysis reference 20 is cited, for which it is mentioned that full text is only available in Chinese. Give complete details of GC-MS methods used in the study in the manuscript

Already revised in the manuscript

---

## [Editor Report · Decision Letter 4]

22 May 2024

Isolation and screening of fungi for enhanced agarwood formation in Aquilaria sinensis trees

PONE-D-23-28830R4

Dear Dr. Liu,

We’re pleased to inform you that your manuscript has been judged scientifically suitable for publication and will be formally accepted for publication once it meets all outstanding technical requirements.

Kind regards,

Niraj Agarwala, Ph.D.

Academic Editor

PLOS ONE
---

## [Editor Report · Acceptance letter]

5 Jun 2024

PONE-D-23-28830R4 

PLOS ONE

Dear Dr. Liu, 

I'm pleased to inform you that your manuscript has been deemed suitable for publication in PLOS ONE. Congratulations! Your manuscript is now being handed over to our production team.

Kind regards, 

on behalf of

Dr. Niraj Agarwala 

Academic Editor

PLOS ONE